# Multiplex CRISPR/Cas screen in regenerating haploid limbs of chimeric Axolotls

Lucas D Sanor[1†], Grant Parker Flowers[1†], Craig M Crews[1,2,3]*

[1]Department of Molecular, Cellular, and Developmental Biology, Yale University, New Haven, United States; [2]Department of Chemistry, Yale University, New Haven, United States; [3]Department of Pharmacology, Yale University, New Haven, United States

**Abstract** Axolotls and other salamanders can regenerate entire limbs after amputation as adults, and much recent effort has sought to identify the molecular programs controlling this process. While targeted mutagenesis approaches like CRISPR/Cas9 now permit gene-level investigation of these mechanisms, genetic screening in the axolotl requires an extensive commitment of time and space. Previously, we quantified CRISPR/Cas9-generated mutations in the limbs of mosaic mutant axolotls before and after regeneration and found that the regenerated limb is a highfidelity replicate of the original limb (Flowers et al. 2017). Here, we circumvent aforementioned genetic screening limitations and present methods for a multiplex CRISPR/Cas9 haploid screen in chimeric axolotls (MuCHaChA), which is a novel platform for haploid genetic screening in animals to identify genes essential for limb regeneration.

*For correspondence:
craig.crews@yale.edu

[†]These authors contributed equally to this work

Competing interests: The authors declare that no competing interests exist.

## Introduction

Salamanders are the only vertebrates known to regenerate complete limbs as adults. The axolotl, a species of salamander, can regenerate limbs, tails, and gills without scarring. Regeneration of these complex structures occurs through the formation of a blastema, a mass of proliferating dedifferentiated cells and pre-existing progenitor and stem cells (*Currie et al., 2016*; *Kragl et al., 2009*; *Sandoval-Guzmán et al., 2014*). Transcriptional profiling of the limb blastema has produced long lists of candidate genes that, to date, remain largely uncharacterized (*Bryant et al., 2017*; *Campbell et al., 2011*; *Gerber et al., 2018*; *Knapp et al., 2013*; *Leigh et al., 2018*; *Monaghan et al., 2009*; *Voss et al., 2015*).

The advent of CRISPR/Cas9 has made the axolotl a genetically tractable organism and the functional interrogation of these genes possible (*Flowers et al., 2014*; *Fei et al., 2014*). Although near-complete knockout F[0] animals can be generated, they appear to be universally mosaic, harboring a variety of mutant alleles (*Flowers et al., 2014*). Such embryonically generated mutations both perturb the function of the targeted gene and uniquely label affected cell lineages with a traceable genetic barcode. Previously, we measured the fidelity of limb regeneration by using next-generation sequencing (NGS) to quantify the mutant allele frequencies of multiple genomic loci before and after limb regeneration in mosaic mutant axolotls (*Flowers et al., 2017*). We found that the majority of very low-frequency alleles reoccur in a regenerated limb at a frequency strikingly similar to that of the original limb. These data indicate that limb regeneration is a high-fidelity process in which the contributions of small cell populations to the original limb are replicated in the regenerated limb (*Figure 1C,D*).

Recent single-cell sequencing of the axolotl limb blastema demonstrated that cell identities converge at a transcriptional level during regeneration (*Gerber et al., 2018*). This suggests a shared

genetic program across most blastemal cells. We anticipated that genetic perturbation of critical blastema-enriched genes would impair mutagenized cells' ability to participate in the regenerative process. Negative selection screens are widely used to identify genes essential for cellular processes with CRISPR/Cas (*Shalem et al., 2014*; *Wang et al., 2014*; *Yin and Chen, 2017*). Screening can be improved by using haploid cells, which harbor a single copy of each gene, and thus require monoallelic inactivation to unveil loss-of-function phenotypes. We sought to determine whether we could detect negative selection of mutant alleles in regenerated haploid limbs of axolotls. (*Figure 1C,D*).

## Results

We generated gynogenetic haploids through in vitro activation of eggs from *white* or transgenic RFP+ females using UV-enucleated sperm from a transgenic GFP+ male (*Figure 1A,B*). Haploidy was confirmed by karyotype (n = 14, 3/3 embryos, three squashes/embryo, *Figure 2—figure supplement 1A*), the universal appearance of the haploid syndrome embryonic phenotype (120/120 embryos, *Figure 2—figure supplement 1B,C*; *Hronowski et al., 1979*), and complete absence of paternally-derived GFP expression in donor embryos (156/156 GFP-, *Figure 2—figure supplement 1B*). Adult haploid axolotls are not viable, so we developed reliable whole limb bud grafting techniques to generate chimeric axolotls with haploid limbs (*Figure 1A*, *Figure 2—figure supplement 1D*). To find the optimal embryonic stage for limb bud grafting, we performed reciprocal grafts between stage-matched *white* and GFP+ diploid embryos across a range of developmental stages

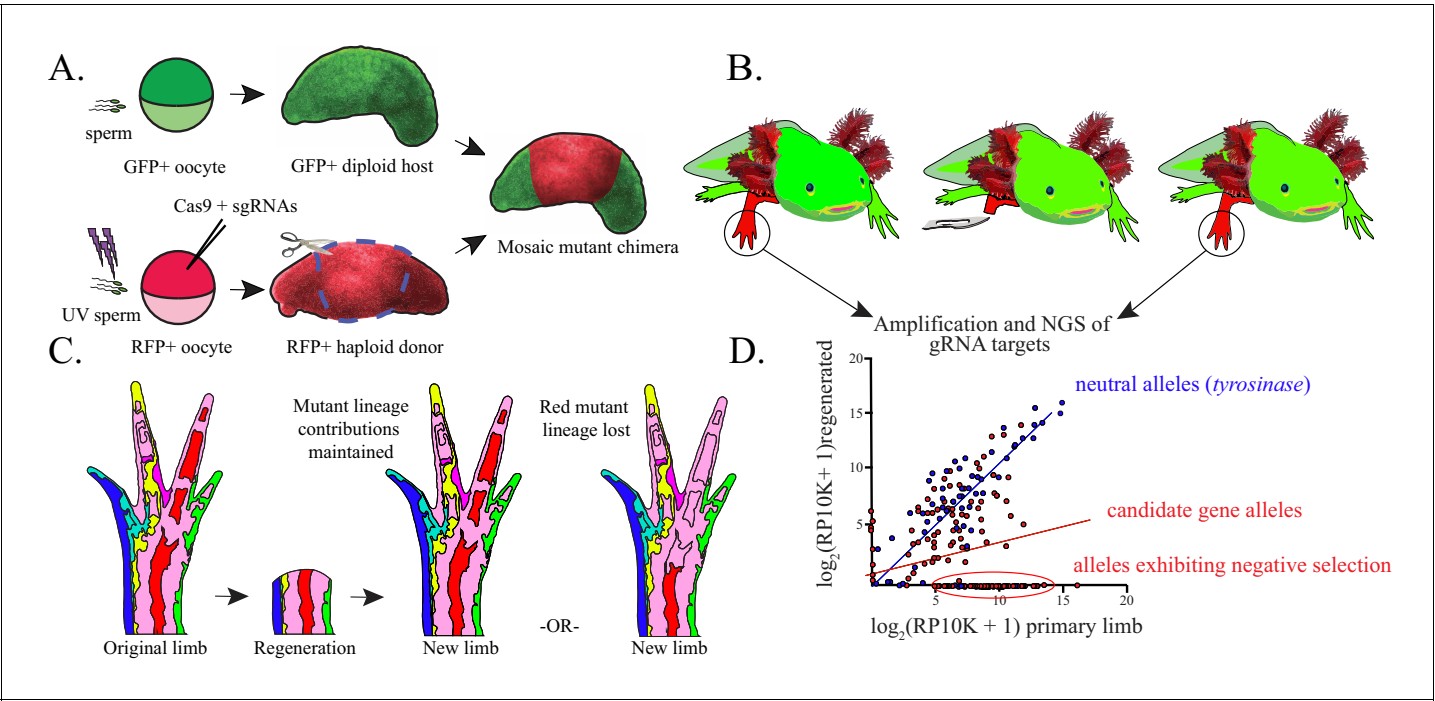

**Figure 1.** Haploid-diploid chimeric generation and lineage analysis. (**A**) Schematic of haploid-diploid chimera generation. Gynogenetic haploids are generated by in vitro activation of unfertilized eggs with UV-enucleated sperm and subsequently mutagenized using CRISPR/Cas9. Chimeric embryos are generated by replacing the limb buds of GFP+ diploid embryos with corresponding tissue from mutagenized haploid donors. (**B**) DNA is extracted from original and fully regenerated haploid limbs of juvenile chimeric axolotls, target sequences are PCR amplified, and these products are subjected to NGS. (**C**) Schematic depicting the contribution of mutant cell lineages to the original and regenerated limb. Cell lineages mutant for non-essential candidate genes (light blue, dark blue, yellow, purple, red, green) may participate normally in regeneration and therefore contribute to the regenerated limb and original limb in a similar proportion. Cell lineages harboring deleterious mutant alleles deleterious (red, far right) are predicted to be reduced in regenerated limbs. (**D**) A hypothetical linear regression plot of the log2 of reads per ten thousand (RP10K+1) of mutant alleles before and after regeneration. Mutant alleles of a neutral gene, tyrosinase (blue), are faithfully preserved between original and regenerated limbs. Mutagenized genes essential for regeneration (red) will show a decrease in allele frequency or a complete loss of alleles in the regenerated limb.

(*Figure 2—source data 1*). Diploid-diploid chimera (DDC) graft limbs were scored for the presence or absence of GFP+ host-derived cells using a fluorescent microscope. Embryos grafted at stage 23–25 produced normally developed limbs with a consistent host-derived neural GFP+ expression pattern (*Figure 2B*; *Figure 2—source data 1*). We adapted the DDC grafting protocol for haploids by substituting diploid tissue with that of haploid donors. We found that cleanly grafted haploid limbs develop fully, but are smaller and shorter than the opposing diploid limbs of the same animals (*Figure 2A*, *Figure 2—figure supplement 2*). Furthermore, haploid-diploid chimeras (HDCs) exhibited a neural-GFP expression pattern similar to DDCs (*Figure 2B*).

Next, we tested the regenerative capacity of HDC and DDC graft limbs. We amputated HDC and DDC limbs and found that both fully regenerate and retain their neural GFP expression pattern (2/2 HDC limbs, 2/2 DDC limbs). While the gross morphology of regenerated haploid limbs is identical to that of the original limbs, haploid limb regeneration is slightly delayed relative to diploid limb regeneration (*Figure 2—figure supplement 2*). To quantify the fidelity of haploid limb regeneration, we generated HDCs using haploid donors mutagenized at one of two genomic loci non-essential for regeneration, *tyrosinase* and *methyltransferase-like,* for which we had previously observed faithful recapitulation of mutant allele frequencies between original and regenerated diploid limbs. NGS of targeted loci in 12 HDCs mutagenized with one of two highly active guide RNAs (gRNAs) revealed 92 total alleles with a mean mutation frequency of 3.46% per allele in the primary limbs (SE = + /- 1.19%). NGS of these targeted sites in DNA from regenerated limbs revealed that the log score of the normalized read numbers for each allele in the primary limbs predicts the log score of the normalized read numbers in the secondary limbs ($R^2$ = 0.544, p<0.0001, *Figure 3A,B*, *Figure 3—figure supplement 1*), which is similar to observations made with these same targets in diploid mosaic limbs (*Flowers et al., 2017*). Thus, with respect to morphology and cell lineage contributions, haploid limb regeneration is similar to that of diploid limb regeneration.

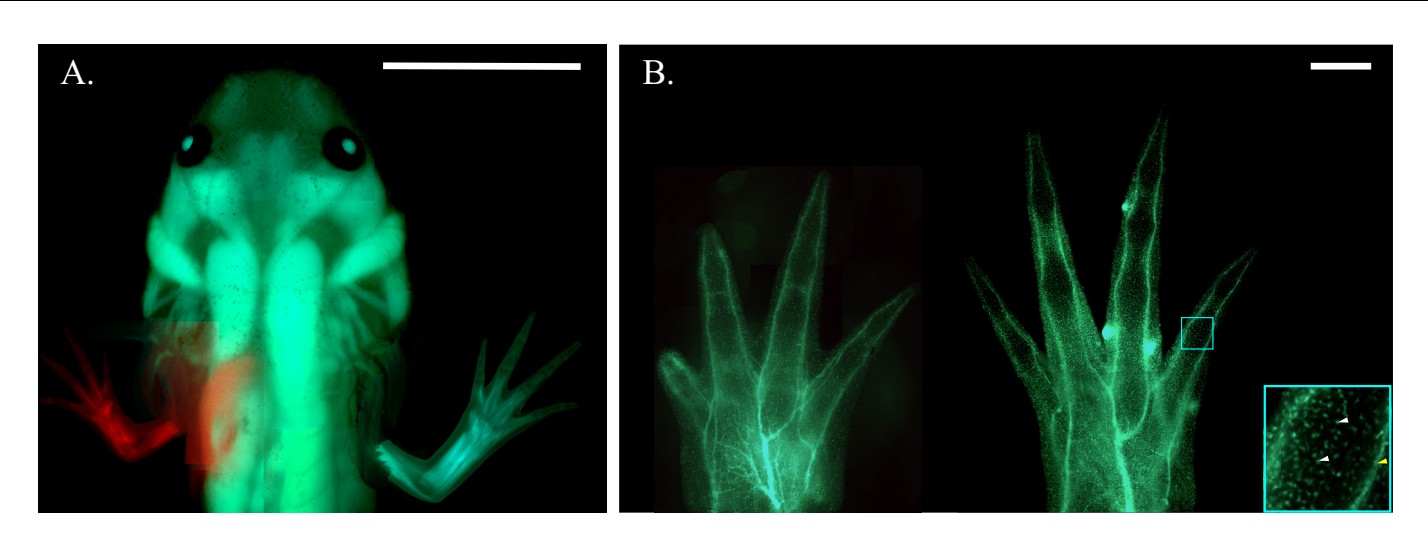

**Figure 2.** Haploid-diploid chimeric axolotl. (**A**) Composite fluorescent image of a chimeric axolotl produced from a limb bud graft from an RFP+ haploid embryo to a GFP+ diploid host. Scale bar = 1 cm. (**B**) Composite fluorescent image of haploid (left) and diploid (right) limbs produced by embryonic limb bud grafting from a *white* donor embryo to a GFP+ diploid host. Both the GFP- haploid limb and GFP- diploid limb grafted to a GFP+ diploid host exhibit a GFP expression pattern that appears to be restricted to spinal nerves innervating the limb (yellow arrow) and individual sensory neurons and blood-derived cells (white arrows) stemming from the host body. Blue box is at 4x magnification (bottom right). Scale bars = 1 mm. Composite images were generated by manually compiling individual photos. Images have been adjusted with cropping, contrast, color correction, and gamma correction.

The online version of this article includes the following source data and figure supplement(s) for figure 2:

**Source data 1.** The number of diploid *white* to diploid GFP+ grafts that were performed to determine the optimal embryonic stage for limb bud grafting.
**Figure supplement 1.** Characterization of haploid larvae.
**Figure supplement 2.** Time course of haploid and diploid limb regeneration.

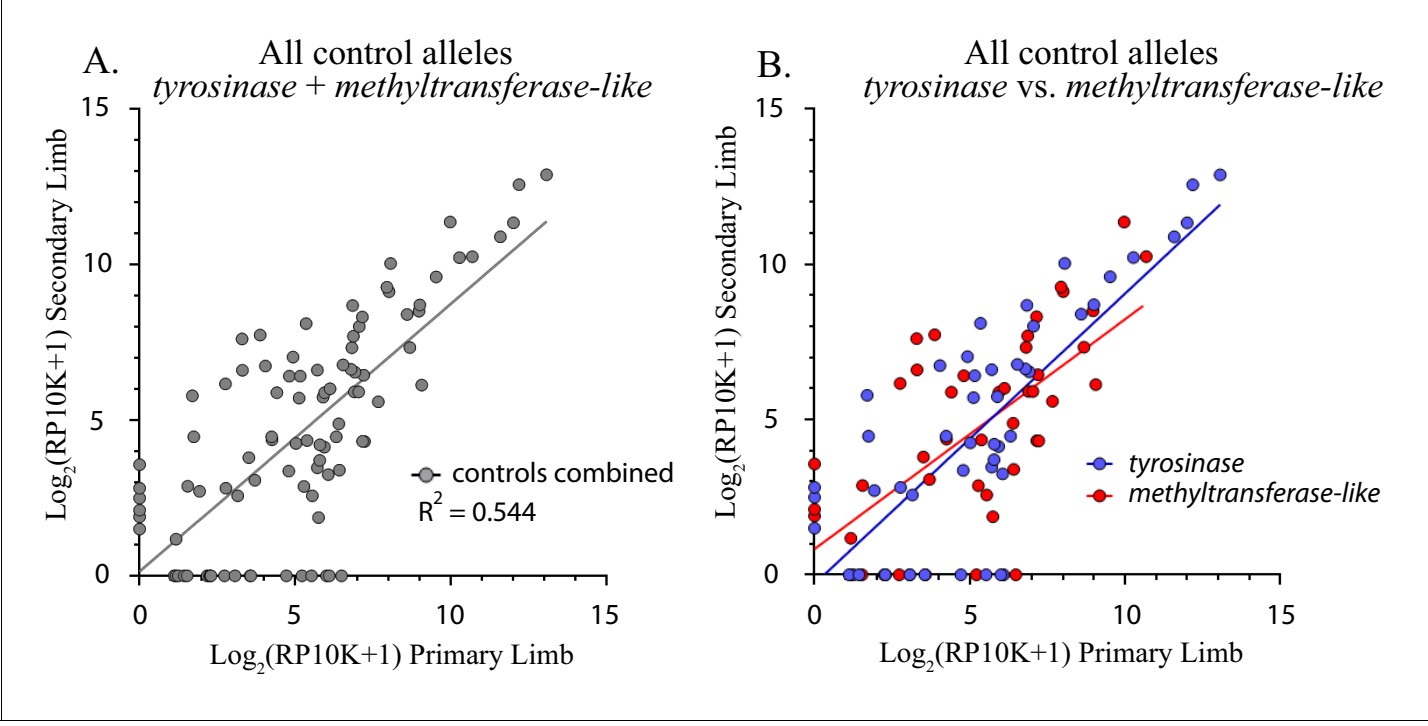

**Figure 3.** Control alleles. (**A**) Comparison of all alleles generated in the controls (*methyltransferase* plus *tyrosinase*) in the original and regenerated haploid limbs of 12 animals. The log scores of the reads per ten thousand (RP10K) of every allele in the original limb are significantly correlated with those of the secondary limb ($R^2$ = 0.544, p-value<0.0001). (**B**) Linear regression comparing the log scores of RP10K for alleles depicted in 3A, but separated by gene (*methyltransferase-like* in red and *tyrosinase* in blue). The slopes of the regression lines are not significantly different for the two genes (*methyltransferase-like m* = 0.740, *tyrosinase m* = 0.935, p-value=0.238, ANCOVA).
The online version of this article includes the following figure supplement(s) for figure 3:

**Figure supplement 1.** Comparison of all alleles generated in the controls (*methyltransferase* and *tyrosinase*) in the original and regenerated haploid limbs of 12 animals shown individually and compared to the entire remaining set of control alleles.

**Figure supplement 2.** Histograms depicting the log of fold change after regeneration for alleles detected in the controls (*methyltransferase-like* and *tyrosinase*).

The majority of *tyrosinase* and *methyltransferase-like* alleles (76.1%, 70/92) are detected in both the first and second haploid limbs. Most mutant alleles occur at a low frequency, comprising fewer than 1.6% of the total reads for a given haploid limb (81.5%, 75/92, *Table 1*). The majority of low-frequency alleles are detected in both primary and secondary limbs (70.7%, 53/75) and undergo less than a two-fold change in frequency after regeneration (69.3%, 52/75, *Table 1*, *Figure 3—figure supplement 2A–D*). Collectively, these results support the notion that, as in diploids, haploid limb regeneration is a high-fidelity process in which the majority of small cell lineages contribute to the regenerated limb in a manner similar to their contributions to the original developed limb.

We found two genes, *fetuin-b* and *catalase*, that exhibited signs of negative selection, showing both a loss of mutant alleles and a decline in the contribution of mutant alleles from primary to secondary limbs (*Table 1*). We compared the linear regression line slopes of all mutant alleles between primary and secondary limbs for each target gene with those of the inessential controls (*methyltransferase-like* and *tyrosinase*) and found that *fetuin-b* (*fetub*) was significantly different (n = 48 mutant alleles, *fetub m* = 0.254, controls *m* = 0.861, p<0.0001, *Figure 4A,C*). Further comparison of *fetub* with all other target genes combined reveals that the slope of the linear regression of *fetub* is lower than that of all other target genes combined (*fetub m = 0.254*, All other target genes *m* = 0.619, p=0.009, ANCOVA, *Figure 4D*). Linear regression analysis of *fetub* reveals that the log scores of the normalized read numbers for each allele in the second limb poorly predict the log scores of the normalized read numbers in the primary limb ($R^2$ = 0.069, p=0.046, *Figure 4A*). Alleles of *fetub* detected in the primary limb are more likely to be absent in the secondary limb (45.8%, 22/48) than

**Table 1.** The numbers of all alleles in the first limbs of controls, all targets, *fetuin-b*, all targets excluding *fetuin-b*, *catalase*, and all targets excluding *catalase* that are sorted by mutation frequency and log of fold change.

**Controls**

| Allele Frequency | | | | Log of fold change | |
|---|---|---|---|---|---|
| (Low) Frequency < 1.6% | | | | < 2 | > 2 |
| Alleles Lost | 22 | | | 5 | 17 |
| Alleles Preserved | 53 | | | 35 | 18 |
| Sum | 75 | | | 40 | 35 |
| Allele Frequency | | | | Log of fold change | |
| Frequency > 1.6% | | | | < 2 | > 2 |
| Alleles Lost | 0 | | | 0 | 0 |
| Alleles Preserved | 17 | | | 15 | 2 |
| Sum | 17 | | | 15 | 2 |
| Total alleles: 92 | | | | | |

**All targets**

| Allele Frequency | | | Log of fold change | |
|---|---|---|---|---|
| (Low) Frequency < 1.6% | | | < 2 | > 2 |
| Alleles Lost | 60 | | 24 | 36 |
| Alleles Preserved | 94 | | 71 | 23 |
| Sum | 154 | | 95 | 59 |
| Allele Frequency | | | Log of fold change | |
| Frequency > 1.6% | | | < 2 | > 2 |
| Alleles Lost | 2 | | 0 | 2 |
| Alleles Preserved | 20 | | 13 | 7 |
| Sum | 22 | | 13 | 9 |
| Total alleles: 176 | | | | |

**fetuin-b**

| Allele Frequency | | | | Log of fold change | |
|---|---|---|---|---|---|
| (Low) Frequency < 1.6% | | | | < 2 | > 2 |
| Alleles Lost | 20 | | | 9 | 11 |
| Alleles Preserved | 25 | | | 21 | 4 |
| Sum | 45 | | | 30 | 15 |
| Allele Frequency | | | | Log of fold change | |
| Frequency > 1.6% | | | | < 2 | > 2 |
| Alleles Lost | 2 | | | 0 | 2 |
| Alleles Preserved | 1 | | | 0 | 1 |
| Sum | 3 | | | 0 | 3 |
| Total alleles: 48 | | | | | |

**All targets except fetuin-b**

| Allele Frequency | | | Log of fold change | |
|---|---|---|---|---|
| (Low) Frequency < 1.6% | | | < 2 | > 2 |
| Alleles Lost | 40 | | 15 | 25 |
| Alleles Preserved | 69 | | 50 | 19 |
| Sum | 109 | | 65 | 44 |
| Allele Frequency | | | Log of fold change | |
| Frequency > 1.6% | | | < 2 | > 2 |
| Alleles Lost | 0 | | 0 | 0 |
| Alleles Preserved | 19 | | 13 | 6 |
| Sum | 19 | | 13 | 6 |
| Total alleles: 128 | | | | |

**catalase**

| Allele Frequency | | | | Log of fold change | |
|---|---|---|---|---|---|
| (Low) Frequency < 1.6% | | | | < 2 | > 2 |
| Alleles Lost | 6 | | | 1 | 5 |
| Alleles Preserved | 1 | | | 1 | 0 |
| Sum | 7 | | | 2 | 5 |
| Allele Frequency | | | | Log of fold change | |
| Frequency > 1.6% | | | | < 2 | > 2 |
| Alleles Lost | 0 | | | 0 | 0 |
| Alleles Preserved | 1 | | | 0 | 1 |
| Sum | 1 | | | 0 | 1 |
| Total alleles: 8 | | | | | |

**All other targets except catalase**

| Allele Frequency | | | Log of fold change | |
|---|---|---|---|---|
| (Low) Frequency < 1.6% | | | < 2 | > 2 |
| Alleles Lost | 54 | | 23 | 31 |
| Alleles Preserved | 93 | | 70 | 23 |
| Sum | 147 | | 93 | 54 |
| Allele Frequency | | | Log of fold change | |
| Frequency > 1.6% | | | < 2 | > 2 |
| Alleles Lost | 2 | | 0 | 2 |
| Alleles Preserved | 19 | | 13 | 6 |
| Sum | 21 | | 13 | 8 |
| Total alleles: 168 | | | | |

alleles detected in controls (23.9%, 22/92) and this difference is significant ($\chi^2$ = 7.03, p=0.008). *Fetub* alleles (45.8%, 22/48) are more likely to be absent from the second limb than alleles of all other targets combined (31.3%, 40/128), but this effect is not significant, except when the other outlier, *catalase*, is excluded ($\chi^2$ = 3.25, p=0.071 and $\chi^2$ = 5.23, p=0.022, respectively).

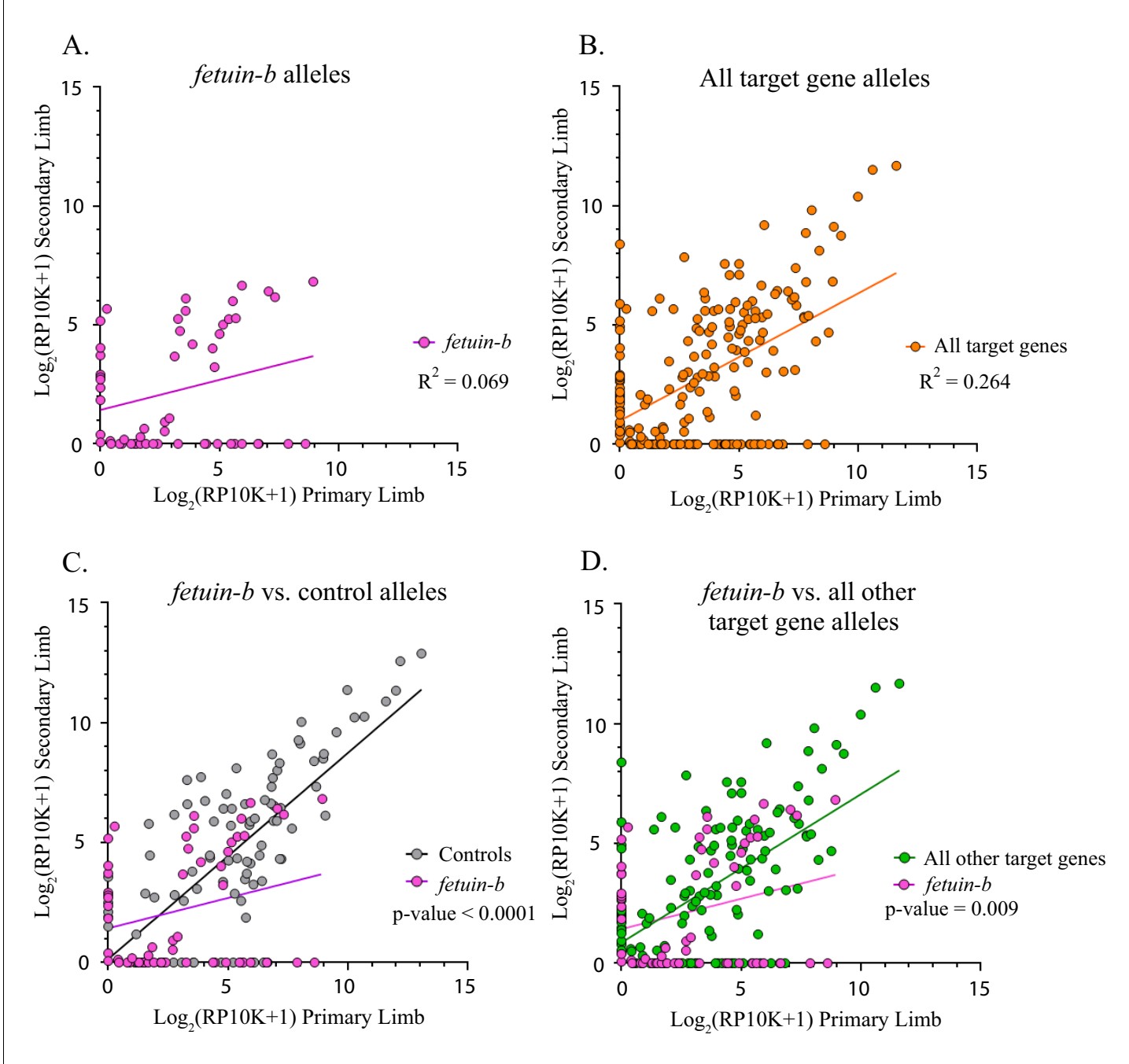

**Figure 4.** Fetuin-b alleles compared to all other target gene and control alleles. (A) Linear regression plot of the $\log_2$(RP10K) score for all alleles of *fetuin-b* detected in the first and regenerated haploid limbs of 11 animals. The log scores of alleles in the primary limb poorly predict the log scores of alleles in the secondary limb. ($R^2 = 0.069$, p-value=0.046). (B) Linear regression plot of the $\log_2$(RP10K) score for all alleles of all targets detected in the primary and regenerated limb ($R^2 = 0.264$, p<0.0001). (C) Comparison of linear regression plots of *fetuin-b* (pink) with controls (gray). The slopes of the regression lines are significantly different (*fetuin-b m* = 0.254, *controls m* = 0.861, p-value<0.0001, ANCOVA). (D) Comparison of linear regression plots of *fetuin-b* (pink) with all other targets (green). The slopes of the regression lines are significantly different (*fetuin-b m* = 0.254, *all other targets m* = 0.619, p-value=0.009, ANCOVA).

The online version of this article includes the following source data and figure supplement(s) for figure 4:

**Source data 1.** Raw number of reads, normalized reads, and $\log_2$(RP10K) score for all mutant alleles of every targeted gene in each mutant limb in this study.

*Figure 4 continued on next page*

*Figure 4 continued*

**Figure supplement 1.** Linear regression plot of the log$_2$(RP10K) score for all alleles in primary and secondary of each targeted gene for which no significant deviation was detected from that of control alleles (*akap8l*, p=0.069; *cacng,* p=0.166; *hnrnpa0*, p=0.371; *hoxa9*, p=0.637; *hoxb13*, p=0.053; *myl6*, p=0.850; *pmp2*, p=0.624; *rcc*, p=0.176; *tyr*, p=0.532,; *zic5*, p=0.480; ANCOVA).

Similarly, the slope of the linear regression of *catalase* alleles differed from control genes (n = 8 mutant alleles, *catalase m* = 0.018, controls *m* = 0.861, p=0.005, ANCOVA, *Figure 5A,C*). The slope of the linear regression of *catalase* did not differ from that of all other target genes combined, except when *fetub* was excluded (*catalase m* = 0.018, all other target genes *m* = 0.550, p=0.073, all other target genes excluding *fetub m* = 0.645, p=0.029, ANCOVA, *Figure 5B,D*). A significantly greater proportion of *catalase* alleles are lost (75.5%, 6/8) than both those of controls and all other targets combined ($\chi^2$ = 9.53, p=0.002 and $\chi^2$ = 5.81, p=0.016, respectively).

To increase the total number of *catalase* and *fetub* mutants analyzed, we next addressed whether loss of these genes perturbs regeneration at a whole organismal level. We produced early embryonic mutants for *catalase, fetub*, and *tyrosinase* by injecting gRNAs against each with Cas9 protein into zygotes. At stage 44, we amputated the posterior 2 mm of the tails of each larva and monitored its regeneration. We extracted DNA from the amputated tails and confirmed the high-level mutagenesis of *fetub* and *catalase* by fluorescent PCR fragment analysis (*fetub*, n = 12, mean = 7.3% wild-type alleles, SD = + /- 5.2%;. *catalase*, n = 16, mean = 3.0% wildtype alleles, SD = + /- 5.8%, *Figure 6—source data 1*). *fetub* and *catalase* mutants did not display regeneration growth defects compared to *tyrosinase* mutants at early time points, but the total regenerative outgrowth of both *fetub* and *catalase* mutant tails were reduced compared to *tyrosinase* mutants at 18 days post-amputation (n = 16 *tyrosinase* mutants, p=0.002, *fetub*; p=0.012, *catalase*; Welch's t-test, one-tailed; *Figure 6A–C*), with the reduction in regeneration also evident at 14 days post-amputation in *catalase* mutant tails (p=0. 025, Welch's t-test, one-tailed, *Figure 6A*). These data indicate that, while *catalase* and *fetub* are not essential for the onset of regeneration, the process of regeneration is slower in the tails of *catalase* and *fetub* mutants. These findings are consistent with the apparent loss of catalase and *fetub* mutant cells within the context of regenerating mosaic mutant haploid limbs and suggest a broader role for these genes in the regeneration of multiple tissues and structures. As cell competition in developing tissues can result in the elimination of cells lacking genes controlling the rate of growth at a whole organismal level (*Johnston et al., 1999*; *Morata and Ripoll, 1975*), these findings support the validity of this assay as a means to identify genes critical for proper regeneration.

## Discussion

Collectively, our data suggests that cells lacking the limb blastema-enriched genes, *fetub* and *catalase*, have a reduced capacity to contribute to the regenerating limb. *Catalase* is an enzyme that plays a conserved role in protecting cells from oxidative damage by catalyzing the decomposition of hydrogen peroxide, a reactive oxygen species (ROS). Despite their potentially harmful effects, ROS are critical for normal tail, fin, and heart regeneration to proceed in *xenopus* and zebrafish (*Love et al., 2013*; *Gauron et al., 2013*; *Han et al., 2014*). However, prolonged ROS-exposure and ROS-induced cellular senescence impair tissue regeneration (*Saxena et al., 2019*). Overexpression of *catalase* impedes heart regeneration after infarction in zebrafish, and chemical inhibition of Catalase may transiently delay tail regeneration in *xenopus* larva, suggesting that ROS levels must be carefully regulated during regeneration (*Han et al., 2014*; *von HAHN, 1959*).

*Fetuin-B* and its paralogue *Fetuin-A,* are highly expressed in the liver, where they are secreted into the blood plasma, and are also expressed in the chondrocytes and muscle cells of developing limb buds in mouse, rat, and sheep (*Terkelsen et al., 1998*; *Saunders et al., 1994*; *Dziegielewska et al., 1996*; *Denecke et al., 2003*). Mammalian Fetuins belong to the cystatin superfamily of proteins, which include many protease inhibitors, yet these two proteins appear to have differing biochemical activities (*Denecke et al., 2003*; *Karmilin et al., 2019*). Fetuin-A is expressed in the growth plate chondrocytes of young mice and is required for proper long bone development, and *Fetuin-A* knockout mice exhibit severely foreshortened femora due to growth plate deformations and displaced distal epiphyses (*Seto et al., 2012*). *Fetub*, knockout mice,

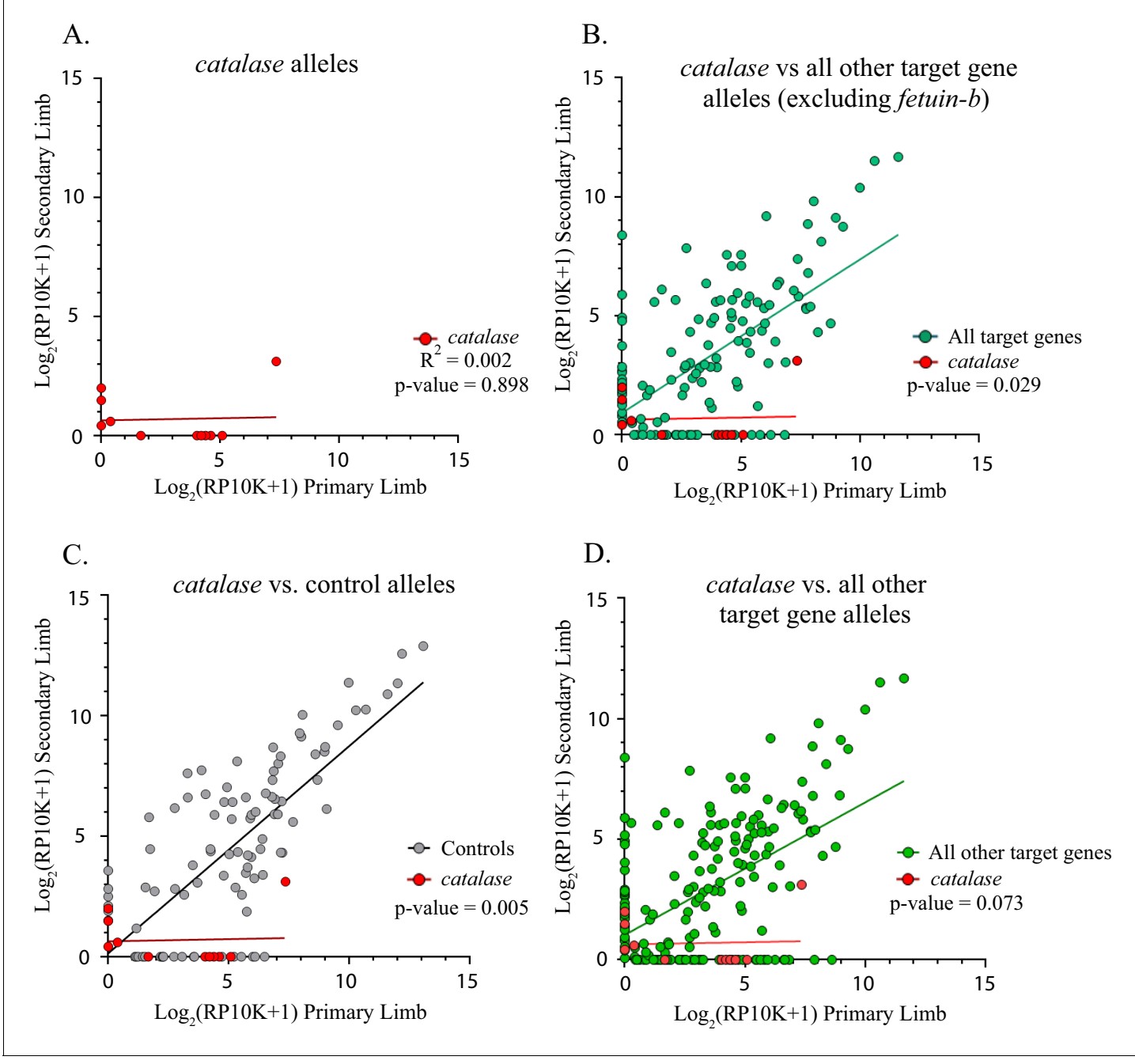

**Figure 5.** Catalase alleles compared to all other target gene and control alleles. (**A**) Linear regression plot of the $\log_2$(RP10K) score for all alleles of *catalase* detected in the first and regenerated haploid limbs of three animals. The log scores of alleles in the primary limbs do not predict the log scores of alleles in the secondary limbs. ($R^2$ = 0.002, p-value=0.898). (**B**) Comparison of linear regression plots of *catalase* (red) with all other targets excluding *fetuin-b* (teal). The slopes of the regression lines are significantly different (*catalase* m = 0.018, all other targets excluding *fetuin-b* m = 0.645, p-value=0.029, ANCOVA). (**C**) Comparison of linear regression plots of *catalase* (red) with controls (gray). The slopes of the regression lines are significantly different (*catalase* m = 0.018, *controls* m = 0.861, p-value=0.005). (**D**) Comparison of linear regression plots of *catalase* (red) with all other targets (green). The slopes of the regression lines are not significantly different (*catalase* m = 0.018, all other targets m = 0.550, p-value=0.073, ANCOVA).

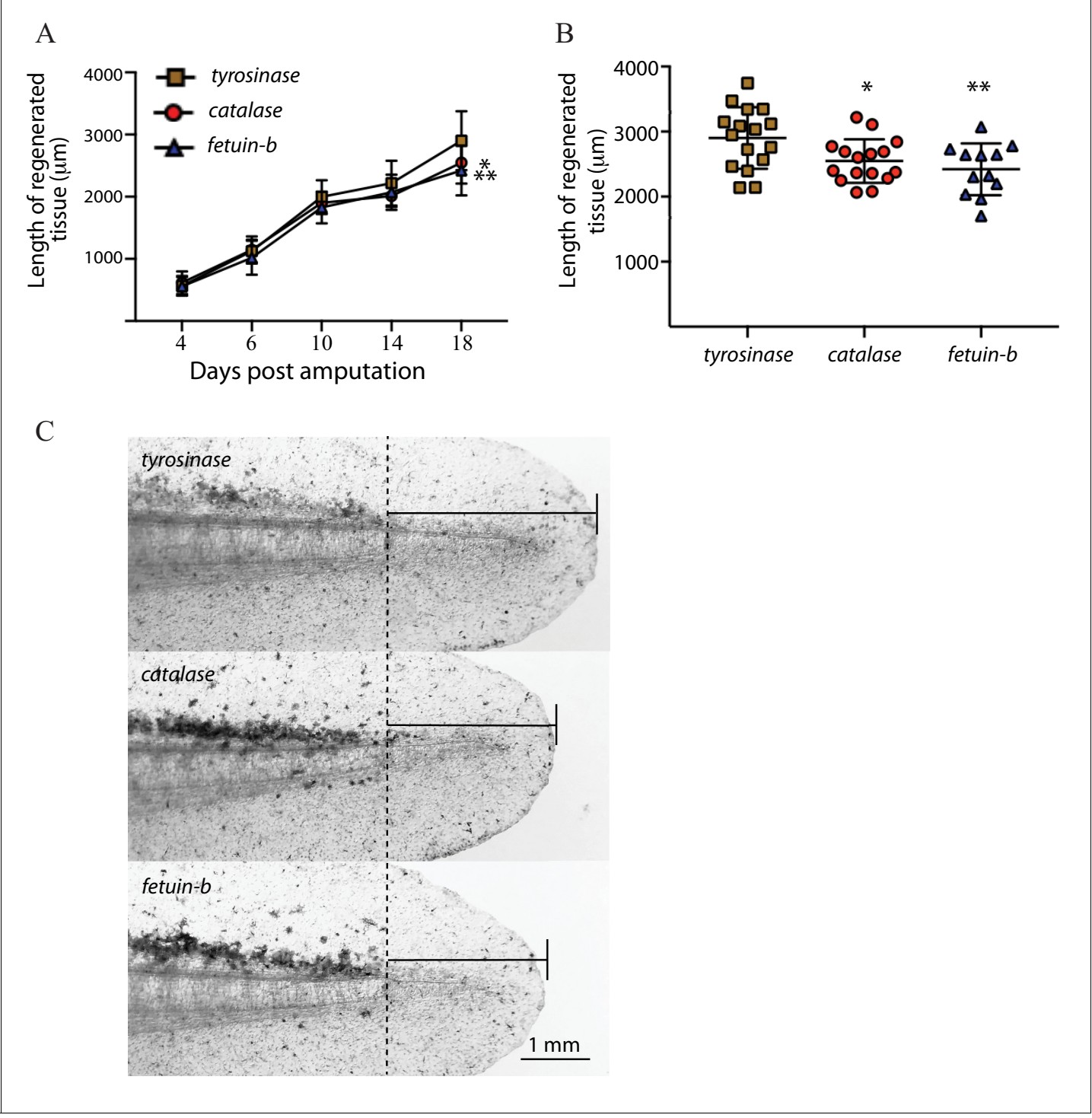

**Figure 6.** Larval tail regeneration in tyrosinase, catalase, and fetuin-b mutants. (**A**) Regenerative outgrowth of tail in high-level *tyrosinase*, *catalase*, and *fetuin-b* $F_0$ mutants. While no significant difference is detected at early time points, both *fetuin-b* and *catalase* mutants display tail reduced tail regeneration compared to *tyrosinase* mutants at later time points (*catalase* vs *tyrosinase*, Day 4, p=0.205, Day 6, p=0.400, Day 10, p=0.111. Day 14, p=0.026, Day 18, p=0.011; *fetuin-b* vs *tyrosinase,* Day 4, p=0.450, Day 6, p=0.129, Day 10, p=0.047, Day 14, p=0.109, Day 18, 0 = 0.002, Welch's t-test). Bars indicate standard deviation. (**B**) Plots of lengths of regenerate in individual *tyrosinase*, *catalase*, and *fetuin-b* $F_0$ mutants at 18 days post-amputation; \*\*=*fetuin* b, p=0.002, \*=*catalase*, p=0.011. (**C**) Brightfield images of individual *tyrosinase*, *catalase*, and *fetuin-b* $F_0$ mutants at 18 days post-amputation (dpa) showing median amount of tail regeneration at 18 dpa. Dotted line indicates the amputation plane.
The online version of this article includes the following source data for figure 6:

**Source data 1.** Regenerative outgrowth measurements and genotyping data for *tyrosinase*, *catalase,* and *fetuin-b* $F_0$ mutants.

however, do not display these defects, and instead show female infertility (*Dietzel et al., 2013*); and mammalian Fetuin-B, unlike Fetuin-A, appears to function as a specific inhibitor of meprin and ovastacin metalloproteinases (*Karmilin et al., 2019*). Extracellular matrix remodeling by metalloproteinases is crucial for a variety of processes, including regeneration. Together, our results suggest that locally expressed Fetub is an important regulator of regeneration in the axolotl.

Both genes for which mutant cells exhibited negative selection in this assay are not developmentally essential in mice; however, transcriptional profiling of axolotl limb blastemas across the time course of regeneration indicates that a considerable portion of blastema-enriched genes are known to participate in limb development or cell survival in other organisms (*Monaghan et al., 2012*; *Knapp et al., 2013*; *Stewart et al., 2013*). We anticipate that mosaic loss-of-function of many genes enriched in both the limb bud and limb blastema may result in a depletion of mutant cell lineages prior to limb formation. Thus, mutant alleles of genes required for both limb development and regeneration may not exhibit negative selection in this assay. Exclusion of mutant alleles prior to limb formation may potentially be investigated by comparing allele frequencies from targeted loci in non-grafted tissues in donor haploid embryos to those found within developed limbs arising from tissue grafts from the same embryos. The identification and validation of *catalase* and *fetub* in this assay suggests that this method is particularly useful for identifying genes that are not essential for limb development but critical for proper regeneration.

Recent single-cell analyses of blastema cells across the time course of regeneration indicate that connective-tissue-derived blastema cells transcriptionally converge to express a shared set of genes distinct from that expressed in developing limb buds; however, at later stages of regeneration these blastema cells recapitulate transcriptional programs found in developing limbs (*Gerber et al., 2018*). These findings indicate that the genes controlling early blastema formation do not substantially overlap with those required for proper limb development. There are several known examples demonstrating dissociation between the genetic control of limb of development and regeneration in the axolotl. The long-characterized recessive short-toes allele in axolotls produces animals that develop limbs but display a progressive decline in regenerative capacity (*Del Rio-Tsonis et al., 1992*; *Gassner and Tassava, 1997*). Similarly, loss of *sox2* in axolotls produces an early spinal cord regeneration defect without perturbing spinal cord development (*Fei et al., 2014*). Nonetheless, because of the potential confounding effects that may arise in this assay when targeting genes essential for limb development, we largely excluded blastema-enriched genes expected to be required for limb development from the set of targeted genes in this study.

Although this negative selection assay in mosaic haploid mutant limbs led to the identification and subsequent confirmation of *catalase* and *fetub* as critical regeneration genes, we failed to detect significant evidence of negative selection of mutant alleles for other assessed candidate genes. For several of these targeted genes, there were insufficient mutant alleles to provide evidence of negative selection (*Figure 4—figure supplement 1*). The absence of significant negative selection for these targets should not be regarded as evidence that these genes are not critical for limb regeneration. As we conducted multiplex mutagenesis to permit analysis of multiple targets in limbs of an individual animal, we reduced the mass of injected gRNAs from that used in earlier CRISPR mutagenesis studies (*Flowers et al., 2017*) to prevent confounding effects caused by mutating multiple targets in single-cell lineages. While 16/25 gRNAs used in this study produced mutant alleles in the limb of at least one animal analyzed, we expect that in future studies injecting greater quantities of gRNA will produce more mutant alleles without confounding results.

Here, we provide a novel screening platform that couples targeted mutagenesis and lineage tracing to identify novel regulators of regeneration. This method relies upon the ability to generate chimeric animals that possess mutagenized haploid limbs. We find that the development and regeneration of these haploid limbs is comparable to that of diploid limbs. Using this approach, we find that *catalase* and *fetuin-b* are required for cells to participate in limb regeneration and for proper tail regeneration. As the axolotl possesses an impressive capacity to regenerate many parts of its body, future work should explore whether haploid chimeric approaches may be applied to the study the regeneration of these structures. To our knowledge, this is the first example of a true in vivo haploid selection screen conducted in a complex structure of a vertebrate.

# Materials and methods

**Key resources table**

| Reagent type (species) or resource | Designation | Source or reference | Identifiers | Additional information |
|---|---|---|---|---|
| Genetic reagent (*Ambystoma mexicanum*) | *cagg:egfp* | Ambystoma Genetic Stock Center (*Sobkow et al., 2006*) | AGSC Cat# 110A, RRID:AGSC_110A | |
| Genetic reagent (*Ambystoma mexicanum*) | *cagg:nls-mcherry* | Ambystoma Genetic Stock Center (*Kragl et al., 2009*) | AGSC Cat# 112A, RRID:AGSC_112A | |
| Chemical compound, drug | MS-222 | Western Chemical | ANADA #200–226 | |
| Chemical compound, drug | Human chorionic gonadotropin (Chorulon,) | Merck Animal Health | NADA 140–927 | |
| Gene (*Ambystoma mexicanum*) | *msx2* | Axolotl transcriptome assembly 3.4 | AMEXTC_0340000067092 | |
| Gene (*Ambystoma mexicanum*) | *prmt1* | Axolotl transcriptome assembly 3.4 | AMEXTC_0340000062704 | |
| Gene (*Ambystoma mexicanum*) | *myl6* | Axolotl transcriptome assembly 3.4 | AMEXTC_0340000067862 | |
| Gene (*Ambystoma mexicanum*) | *fetub* | Axolotl transcriptome assembly 3.4 | AMEXTC_0340000227254 | |
| Gene (*Ambystoma mexicanum*) | *hoxc8* | Axolotl transcriptome assembly 3.4 | AMEXTC_0340000065333 | |
| Gene (*Ambystoma mexicanum*) | *akap8l* | Axolotl transcriptome assembly 3.4 | AMEXTC_0340000192860 | |
| Gene (*Ambystoma mexicanum*) | *hrnrpa0* | Axolotl transcriptome assembly 3.4 | AMEXTC_0340000081837 | |
| Gene (*Ambystoma mexicanum*) | *hsd17b10* | Axolotl transcriptome assembly 3.4 | AMEXTC_0340000257015 | |
| Gene (*Ambystoma mexicanum*) | *hoxb9* | Axolotl transcriptome assembly 3.4 | AMEXTC_0340000035333 | |
| Gene (*Ambystoma mexicanum*) | *tyrosinase* | Axolotl transcriptome assembly 3.4 | AMEXTC_0340000179254 | |
| Gene (*Ambystoma mexicanum*) | *etv4* | Axolotl transcriptome assembly 3.4 | AMEXTC_0340000233035 | |
| Gene (*Ambystoma mexicanum*) | *cacng1* | Axolotl transcriptome assembly 3.4 | AMEXTC_0340000081988 | |
| Gene (*Ambystoma mexicanum*) | *catalase* | Axolotl transcriptome assembly 3.4 | AMEXTC_0340000186723 | |
| Gene (*Ambystoma mexicanum*) | *hoxb13* | Axolotl transcriptome assembly 3.4 | AMEXTC_0340000007929 | |
| Gene (*Ambystoma mexicanum*) | *zic5* | Axolotl transcriptome assembly 3.4 | AMEXTC_0340000057641 | |
| Gene (*Ambystoma mexicanum*) | *ecm1* | Axolotl transcriptome assembly 3.4 | AMEXTC_0340000123229 | |
| Gene (*Ambystoma mexicanum*) | *cornifelin* | Axolotl transcriptome assembly 3.4 | AMEXTC_0340000173184 | |
| Gene (*Ambystoma mexicanum*) | *dsg-like* | Axolotl transcriptome assembly 3.4 | AMEXTC_0340000056512 | |
| Gene (*Ambystoma mexicanum*) | *enpp2* | Axolotl transcriptome assembly 3.4 | AMEXTC_0340000217071 | |
| Gene (*Ambystoma mexicanum*) | *fabp2* | Axolotl transcriptome assembly 3.4 | AMEXTC_0340000084459 | |

*Continued on next page*

*Continued*

| Reagent type (species) or resource | Designation | Source or reference | Identifiers | Additional information |
|---|---|---|---|---|
| Gene (*Ambystoma mexicanum*) | *pmp2* | Axolotl transcriptome assembly 3.4 | AMEXTC_ 0340000238807 | |
| Gene (*Ambystoma mexicanum*) | *kcne1* | Axolotl transcriptome assembly 3.4 | AMEXTC_ 0340000121776 | |
| Gene (*Ambystoma mexicanum*) | *krt6a* | Axolotl transcriptome assembly 3.4 | AMEXTC_ 0340000060835 | |
| Gene (*Ambystoma mexicanum*) | *rcc1* | Axolotl transcriptome assembly 3.4 | AMEXTC_ 0340000210022 | |
| Recombinant DNA reagent | MLM3613 | (*Hwang et al., 2013*) | RRID: Addgene plasmid 42251 | *Cas9* expression vector |
| Peptide, recombinant protein | Cas9 | PNABio | Cat. #: CP04-500 | |
| Commercial assay or kit | mMessage mMachine Kit | ThermoFisher | Cat. #: Am1345 | |
| Commercial assay or kit | MAXIscript SP6/T7 Transcription Kit | ThermoFisher | Cat. #: Am1322 | |
| Chemical compound, drug | MS-222 | Sigma Aldrich | SML1656 | |
| Software, algorithm | Geneious Software | Biomatters | RRID:SCR_010519 | |

## Animals

All animal experiments were carried out on *Ambystoma mexicanum* (axolotls) in facilities at Yale University. Experimental procedures were approved by the Yale University IACUC (2017–10557) and were in accordance with all federal policies and guidelines governing the use of vertebrate animals. All axolotls used in this study were produced by natural mating or in vitro fertilization and housed in our facility. They were fed artemia, blood worms, and fish pellets. The parental *cagg:egfp* and *cagg: nls-mcherry* transgenic animals were originally obtained from the Ambystoma Genetic Stock Center (*Kragl et al., 2009*; *Sobkow et al., 2006*).

## Haploid generation

Gametes were collected in a manner similar to that previously described (*Mansour et al., 2011*), and haploids were generated using in vitro activation methods (*Trottier and Armstrong, 1976*). RFP+ or white mutant females were anesthetized in 1 g/L MS-222 (Tricaine-S, Western Chemical) and injected with 1500 units of human chorionic gonadotropin (Chorulon, Merck Animal Health) in the dorsal musculature above the hind limbs. Females were stored at 8°C to 10°C for 2 days until they began laying unfertilized eggs. Gametes were then collected from a GFP+ male by gentle pelvic squeezing using a P1000 micropipette. Sperm viability and concentration was assessed using an inverted microscope. Sperm was diluted to 80,000 motile cells/mL in sterile 0.1x MMR and spread on a sterile petri dish to form a 1-mm deep film. Eggs were extracted from the female after full anesthesia in a similar manner, without the use of a pipette. To enucleate, sperm were placed 4 cm from the bulbs of a 254 nm CL-1000 Ultraviolet Crosslinker (UVP) and irradiated with 800,000 uJ/mm$^2$ of UV energy. Each unfertilized egg was then coated with 0.25 to 0.5 µL of enucleated sperm and allowed to sit at room temperature for 30 min. Eggs were then flooded with sterile 0.1x MMR. Haploid embryos were individually housed and maintained from 10°C to 18°C. Haploids were inspected for GFP expression 3 to 4 days after in vitro activation.

## Karyotype analysis

Stage 35 embryos selected for karyotype analysis were staged according to the Schreckenberg and Jacobson staging series (*Schreckenberg and Jacobson, 1975*). Embryos were prepared for karyotype analysis by incubating them in 0.25% colchicine in 0.1x MMR for 48 hr at 18°. Embryos were then washed twice with dissociation solution (0.1x MMR without Ca$^{2+}$ or Mg$^{2+}$) with 0.25%

Colchicine. Using fine forceps, the ventral half of each embryo was removed and individually transferred to a microcentrifuge tube containing 1 mL of dissociation solution and incubated at room temperature for 5 min. Using a Pasteur pipette, cells were loosened by gentle agitation and allowed to settle. All but 50 µL of the dissociation solution was removed without disturbing the cells and replaced with 950 L of 60% acetic acid in water, gently mixed, and allowed to stand for 5 min. The fixed cells were pipetted onto positively charged slides, which were briefly flamed to dry before a cover slip was added. Without disturbing the coverslip, a 50-lb lead brick with a paper towel cushion was placed on the slide to squash the cells. After 5 min, the brick was removed, the slides frozen on dry ice, and the coverslips were pried off with a scalpel. The samples were then stained with Hoechst 33342, covered, and sealed with clear nail polish.

## Genome editing

Target gRNA sequences are listed in the *Supplementary file 1*. Axolotl matings and microinjections were carried out as previously described (*Flowers and Crews, 2015*) with the following modifications for haploid mutagenesis: Haploids were allowed to develop for 7 hr until they reached a two-cell stage. For multiplex mutagenesis, each blastomere was injected with an equal volume of injection mix for a total of 1000 pg of *cas9* mRNA and 5 pg per gRNA (five total). Each gRNA was injected in two separate pools of gRNAs into embryos from two separate matings. Control embryos were injected as described, but with 50 pg of gRNA. The data described represents the results from ten independent in vitro fertilization, injection, and grafting experiments to produce experimental animals, and three additional experiments to produce control animals.

For tail regeneration experiments, embryos were produced by a single mating and injected with 1250 pg of gRNA coupled with 1250 pg of Cas9 protein (PNA Bio) as described previously within 2 hr of being laid (*Fei et al., 2018*). Successful mutagenesis of *tyrosinase* was assessed by loss of pigmentation. Mutations of *catalase* and *fetuin-b* were confirmed by fragment analysis PCR (*Figure 6—source data 1*).

## Limb field grafting

Stage-matched haploid embryos and GFP+ diploid hosts (stages 21–25) were freed from their vitelline membranes and washed with sterile 0.1x MMR with antibiotics and stored overnight at 4°C. The embryos were then transferred to refrigerated, sterile agar plates with holding grooves and maintained at 8°C on a chilled-stage dissecting microscope. Two sets of Dumostar forceps (Dumont #55, Fine Science Tools) were used to replace the limb bud fields from the GFP hosts with the corresponding tissue from haploid donors. Tissue grafts were held in place for one hour using a small rectangular glass shard from a crushed cover slip. The glass shards were then carefully removed, and the embryos were gently transferred to individual housing at 12°C for 24 hr. Afterwards, the embryos were transferred to new sterile 0.1x MMR with antibiotics and maintained at 16°C to 18°C. 0.1x MMR was replaced every day until the tadpoles began to feed. After 2 to 3 months of development, limbs were inspected for purity and quality using a fluorescent dissecting microscope.

## Amputations

Amputations were performed through the mid-zeugopod as previously described (*Flowers et al., 2017*). Animals were anesthetized in 1 g/L MS-222 (Tricaine-S, Western Chemical). For each gRNA target pool, an additional amputation was performed on a non-mutant animal to serve as a non-mutant control for sequencing. Limbs were frozen at −20° C until all primary and secondary limbs were collected.

Tail amputations were carried out in stage 44 larvae (*Nye et al., 2003*). Larvae were maintained at 19.5°C in 0.1x MMR. Embryos were monitored and imaged with a Zeiss stereomicroscope using brightfield imaging.

## Genomic DNA preparation

Genomic DNA was extracted from entire amputated limbs using the DNeasy Blood and Tissue Collection Kit (Qiagen) according to the manufacturer's protocol with the following modifications: limbs were suspended in a 3x volume and completely digested with vortexing in Proteinase K for 6 to 8 hr at 56°C. Before adding AL buffer and ethanol, the digest was split into three separate tubes of equal

volume for each limb and treated as a separate sample. After the addition of AL buffer and ethanol, the sample was vortexed and frozen at −20℃ overnight. After thawing, the purification process was completed according to the manufacturer's instructions. For tail amputations, DNA was extracted from 2 mm tissue samples using the QiaAmp DNA Kit (Qiagen) according to the manufacturer's instructions.

## Target amplification and genotyping

All PCRs for primary and secondary limbs were performed at the same time using Phusion High-Fidelty DNA Polymerase (NEB). Target sequences were labeled using a three-cycle labeling step with a unique molecular identifier (UMI), purified, and amplified with final 30-cycle PCR using the purified UMI-labeled product as a template. The final PCR product was purified to remove all primers. For the three-cycle UMI labeling step, 500 ng of genomic DNA was used as template in a 100 µL reaction. For each 30-cycle PCR, an indexed universal adaptor forward primer and a gene-specific reverse primer was used. From 5' to 3', each UMI consists of a universal adaptor sequence, a 10N randomized barcode, and 18 to 28 bases of gene specific sequence. Each universal adaptor primer possesses a unique 4 to 6 bp 5' indexing sequence. For each target, this process was carried out for a no-gDNA control. All products were purified using 1.0x Ampure XP magnetic beads (Agencourt). Products were prepared for sequencing using the TruSeq Nano DNA Library Prep Kit and indexed with the Unique Dual Indexes for TruSeq (Illumina). All universal and gene specific primer sequences are listed in *Supplementary file 1*.

The mutation rate in individual larvae for tail regeneration experiments was determined using fragment analysis of fluorescent PCR products as described previously (*Flowers and Crews, 2015*) and analyzed using GeneMapper software. (Thermo Fisher Scientific). Animals in which PCR fragments corresponding to the expected size of wildtype PCR products represented more than 15% of the total intensity of the sum of all PCR products were excluded from analysis.

## Amplicon library QC

To identify and size the various amplicon products in each sample, libraries were analyzed on the Agilent TapeStation 4200 (Agilent Technologies) using the DNA D1000 High Sensitivity assay. Target peaks were identified within the range of 150 bp – 500 bp and samples were then processed for upper and lower automated size selection using the Pippin Prep 2% Agarose, dye-free cassette (Sage Science). Size selected eluates were purified using the Qiagen PCR purification kit (Qiagen) and re-analyzed on the Agilent Tapestation DNA D1000 High Sensitivity assay to confirm the removal of products less than 150 bp and greater than 500 bp. Amplicon libraries were then quantified and normalized using the dsDNA High Sensitivity Assay for Qubit 3.0 (Life Technologies).

## Illumina MiSeq sequencing

The sample library pools were prepared for Illumina MiSeq sequencing following the denaturing and dilution protocol set forth by the manufacturer (Illumina). Prior to MiSeq v2 500 cycle sequencing, the prepared pool was first quality checked on a MiSeq Reagent kit v2 Nano to ensure proper representation of each sample (biased pooling percentage). Deep sequencing on the MiSeq Reagent kit v2 500 cycle was performed with paired end 250 bp reads across five sequencing runs.

## Sequence analysis

Alleles were quantified as previously described (*Flowers et al., 2017*). UMI sequences were extracted from each sequence read after allele assignment, and duplicate sequences were eliminated using the 'eliminate duplicate reads' function in Geneious R11. Based upon the input molecular weight of DNA used in our initial PCRs, we estimated that the sequence reads for each gene for each limb represents a sample of approximately 10,000 cells. We produced normalization coefficients by dividing 10,000 by the total number of unique assigned reads for each target for each animal. The number of unique reads for each allele was multiplied by this coefficient to produce a reads per 10 k (RP10K) value. To obtain coherent log scores, one was added to each value prior to log calculations. All allele sequences, raw allele counts, normalized read numbers, and log values for every animal for each gene can be found in *Figure 4—source data 1*.

## Statistical analyses

The log scores of normalized sequence reads for all alleles of control genes and experimental genes were compared between primary and secondary limbs. Linear regressions were created for each target and control gene, and the slope of each experimental gene regression was compared both to that of all control genes and that of all other experimental genes by one-way ANCOVA in GraphPad Prism software. As two genes were found to differ significantly from control genes, the alleles for those genes were excluded from the set of experimental genes, and the slope of each experimental gene regression was compared to the trimmed set of all other experimental genes.

## Additional information

### Funding

| Funder | Grant reference number | Author |
|---|---|---|
| Connecticut Innovations | Seed Grant 15RMA-YALE-09 | Grant Parker Flowers |
| Eunice Kennedy Shriver National Institute of Child Health and Human Development | Individual Postdoctoral Fellowship F32HD086942 | Grant Parker Flowers |
| Connecticut Innovations | Established Investigator Award 15-RMB-YALE-01 | Craig M Crews |
| National Institute of General Medical Sciences | Predoctoral Training Fellowship T32GM007499 | Lucas D Sanor |

The funders had no role in study design, data collection and interpretation, or the decision to submit the work for publication.

### Author contributions

Lucas D Sanor, Conceptualization, Data curation, Formal analysis, Investigation, Methodology, Writing—original draft, Writing—review and editing; Grant Parker Flowers, Conceptualization, Data curation, Software, Formal analysis, Funding acquisition, Investigation, Methodology, Writing—original draft, Writing—review and editing; Craig M Crews, Conceptualization, Supervision, Funding acquisition, Project administration, Writing—review and editing

### Author ORCIDs

Lucas D Sanor https://orcid.org/0000-0003-2756-0457
Grant Parker Flowers https://orcid.org/0000-0001-7436-3531
Craig M Crews https://orcid.org/0000-0002-8456-2005

### Ethics

Animal experimentation: Experimental procedures were approved by the Yale University IACUC (2017-10557) and were in accordance with all federal policies and guidelines governing the use of vertebrate animals.

### Decision letter and Author response

Decision letter https://doi.org/10.7554/eLife.48511.sa1
Author response https://doi.org/10.7554/eLife.48511.sa2

## Additional files

### Supplementary files

- Supplementary file 1. Primer and gRNA sequences used in study.
- Transparent reporting form

## Data availability

All data generated or analyzed during this study are included in the manuscript or supporting files.

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
