## [Decision Letter]

**Acceptance summary:**

The work provides what we understand to be the first attempt towards a saturation haploid CRISPR/Cas9-based screen for mutations affecting limb/tail regeneration in axolotyl. In limb regeneration, the work has identified guild RNAs that are statistically underrepresented in engrafted tissue regenerates, and highlighted roles for catalase and fetuin-b, confirmed in a tail regenerative assay. While there are clearly limitations to the approach given its statistical basis and potentially mild phenotypes, we appreciate the effort and rigor that has brought the system to this point, and the study should represent a landmark in this field. The authors have considerably improved the manuscript and provided an independent analysis of CRISPR-generated F0 mutations in catalase and fetuin-b compared to tyrosinase controls on tail regeneration in a time course manner, finding delays not in the onset, but in progression of the regenerative process. The study provides a platform for analysis of a broader range of mutants in greater detail and for improvements in approaches with respect to methods and scale in the axolotl system, and should provide a key stimulus for others to attempt analogous screens in other regenerative systems.

**Decision letter after peer review:**

Thank you for submitting your article "Multiplex CRISPR/Cas screen in regenerating haploid limbs of chimeric Axolotls" for consideration by *eLife*. Your article has been reviewed by two peer reviewers, and the evaluation has been overseen by a Reviewing Editor and Patricia Wittkopp as the Senior Editor. The reviewers have opted to remain anonymous.

The reviewers have discussed the reviews with one another and the Reviewing Editor has drafted this decision to help you prepare a revised submission.

The reviewers appreciated the elegance and thoughtfulness of the approach, and the preliminary data generated. The manuscript describes the first application of haploid screening in an animal model. They felt the work potentially provides an important roadmap for relatively rapid limb-specific functional analysis of genes during regeneration using CRISPR – something that has been sorely missing in the field.

However, the following points were raised.

1) Cells with essential regeneration genes knocked out may also be at a disadvantage in the primary transplant to initially generate chimeric animals. This is a limitation and is not discussed. It seems one would want to first target chimeric embryos, then isolate guides from limb bud grafts pre transplant and compare these guide signals to post transplant primary limbs. This would first identify genes that are essential for drop out during original limb formation. Then compare this to secondary limb to define regeneration-specific targets. Examples of where primary and regenerative limb development are under fundamentally different molecular regulation are known and should be highlighted, but in general there may be strong overlap in developmental and regenerative pathways. If so, these targets can be used as controls to establish the power and rigor of the system.

2) Inclusion of control targets known to be essential for regeneration is critical to establish the optimal design for dropout screening in this system, and in our opinion this needs to be done before publication. With the method relying on statistical differences between recovered alleles in primary and secondary limbs, and the very apparent noise in this system, one wants to see a solid positive control essential for regeneration showing significant drop out in pools.

3) In this study, the authors identified fetuin-b and catalase as likely undergoing negative selection but the evidence is only suggestive. To prove the power of this system, it is essential that the authors confirm these two targets are actually required in a confirmation experiment. This would be straightforward if the screen were done just with those targets. Evidence exists for a role of catalase (and ROS) in developing and regenerating organs, and the phenotypes may relate more to differential proliferation in different limb tissues which may translate to allele dropout. Based on previous work in axolotl, 100% targeting. should be possible, notwithstanding the issues of heterogeneous alleles including those introducing in-frame mutations and issue that may arise due to mosaic mutations in non-cell autonomous genes. If these two genes are essential for regeneration, and haploid KO limbs do not regenerate, then that would establish the power of this system. If they are not essential, then this shows the system needs further improvement.

Summary:

Overall there is support for this manuscript, however evidence for the technique as a compelling platform is still missing.

Essential revisions:

For the manuscript to proceed, it will be essential to provide stronger evidence for the above – the inclusion of positive controls that end up being compellingly championed by the method and/or follow up experiments that compellingly show how candidates affect limb regeneration. We appreciate the effort involved in this. The manuscript is not sufficiently self-critical with respect to the method's strengths and limitations. This must also be addressed.

---

## [Author Response]

The reviewers appreciated the elegance and thoughtfulness of the approach, and the preliminary data generated. The manuscript describes the first application of haploid screening in an animal model. They felt the work potentially provides an important roadmap for relatively rapid limb-specific functional analysis of genes during regeneration using CRISPR – something that has been sorely missing in the field.However, the following points were raised.1) Cells with essential regeneration genes knocked out may also be at a disadvantage in the primary transplant to initially generate chimeric animals. This is a limitation and is not discussed. It seems one would want to first target chimeric embryos, then isolate guides from limb bud grafts pre transplant and compare these guide signals to post transplant primary limbs. This would first identify genes that are essential for drop out during original limb formation. Then compare this to secondary limb to define regeneration-specific targets. Examples of where primary and regenerative limb development are under fundamentally different molecular regulation are known and should be highlighted, but in general there may be strong overlap in developmental and regenerative pathways. If so, these targets can be used as controls to establish the power and rigor of the system.

We agree with the reviewers’ point that cells lacking genes essential for regeneration may also be at a disadvantage during limb development, and, thus, we may expect to see a discrepancy between allele frequencies in limb buds and mature limbs. While the time permitted for revisions was not sufficient to explore this experimentally, we have emphasized the likelihood of negative selection during limb development and suggested ways in which this may be explored. We have discussed examples in which there is a dissociation between genes essential for development and regeneration. We have cited recent single-cell transcriptomic work indicating that, prior to recapitulating developmental gene expression, blastema cells exhibit a shared transcriptional profile distinct from that of developing limbs. For these reasons, we believe that this assay is particularly useful for detecting those genes not essential for limb development but critical for proper regeneration.

2) Inclusion of control targets known to be essential for regeneration is critical to establish the optimal design for dropout screening in this system, and in our opinion this needs to be done before publication. With the method relying on statistical differences between recovered alleles in primary and secondary limbs, and the very apparent noise in this system, one wants to see a solid positive control essential for regeneration showing significant drop out in pools.3) In this study, the authors identified fetuin-b and catalase as likely undergoing negative selection but the evidence is only suggestive. To prove the power of this system, it is essential that the authors confirm these two targets are actually required in a confirmation experiment. This would be straightforward if the screen were done just with those targets. Evidence exists for a role of catalase (and ROS) in developing and regenerating organs, and the phenotypes may relate more to differential proliferation in different limb tissues which may translate to allele dropout. Based on previous work in axolotl, 100% targeting. should be possible, notwithstanding the issues of heterogeneous alleles including those introducing in-frame mutations and issue that may arise due to mosaic mutations in non-cell autonomous genes. If these two genes are essential for regeneration, and haploid KO limbs do not regenerate, then that would establish the power of this system. If they are not essential, then this shows the system needs further improvement.

As agreed upon in our previous correspondence with the editor, we generated high-mutation-rate F0 animals with Cas9 protein, amputated their larval tail tips, and monitored tail regeneration. We genotyped these *fetuin-b* and *catalase* mutants to identify those in which mutation rates at the targeted sites were greater than 80% and compared their regeneration to that of control siblings in which the control gene *tyrosinase* had been targeted (loss of *tyrosinase* provided a visual means to confirm successful mutagenesis). We found that *fetuin-b* and *catalase* mutants have impaired tail regeneration, as their tail tips were significantly shorter than that of their *tyrosinase-*mutant siblings. In the absence of well-characterized control genes that could be used for this study, the confirmation of regeneration defects in mutants for both genes that exhibited significant negative selection in regenerated haploid limbs supports the validity of the described approach.